# Organization of the Subdiaphragmatic Vagus Nerve and Its Connection with the Celiac Plexus and the Ovaries in the Female Rat

**DOI:** 10.3390/brainsci13071032

**Published:** 2023-07-06

**Authors:** María E. Rivera-Castro, César F. Pastelín, Juan Bravo-Benítez, Carolina Morán

**Affiliations:** 1Doctorado en Investigaciones Cerebrales, Instituto de Investigaciones Cerebrales, Universidad Veracruzana, Xalapa, Veracruz 91190, Mexico; 2Centro de Investigación en Fisicoquímica de Materiales, Instituto de Ciencias, Benemérita Universidad Autónoma de Puebla, Puebla, Puebla 72960, Mexico; juan.bravob@correo.buap.mx; 3Facultad de Medicina Veterinaria y Zootecnia, Benemérita Universidad Autónoma de Puebla, Tecamachalco, Puebla 75460, Mexico; cesar.pastelin@correo.buap.mx

**Keywords:** vagus nerve, ovary, celiac ganglia, catecholamines, acetylcholine, serotonin

## Abstract

Communication between the ovaries and the central nervous system occurs by peripheral innervation through the celiac plexus, superior ovarian nerve, and ovarian plexus nerve. The vagus nerve is involved in regulating the ovaries, but the neuroanatomical pathway that links them is not clear. Adult female rats were used for gross anatomy, acetylcholinesterase histochemistry, and the immunofluorescence analysis of tyrosine hydroxylase (TH), choline acetyltransferase (ChAT), and tryptophan hydroxylase 2 (TPH). The results obtained indicate that the right vagus nerve (RVN) travels parallel and caudal to the esophagus, where three nerve branches were identified. Also, a right vagal plexus (RVP) formed by microganglia was described, establishing communication with the celiac plexus, and was mainly reactive to tyrosine hydroxylase (TH); some serotoninergic and cholinergic neurons were also found. The left vagus nerve (LVN) travels over the esophagus, bifurcates before its insertion into the stomach and enters the RCG. This neuroanatomical and biochemical description of the RVN and LVN in the rat suggests the RVP is formed by presynaptic catecholaminergic terminals and cholinergic neurons. This information could support detailed studies of communication between the vagus nerve and the ovaries and identify the type of neural signaling involved in abdominal control of the vagus nerve.

## 1. Introduction

The communication of the peripheral and central nervous systems with the ovaries has been studied more closely in the last three decades. It is known that the ovaries in rats receive sympathetic innervation through the superior ovarian nerve (SON) and the ovarian plexus nerve (OPN) [1,2], whereas parasympathetic and sensorial innervation are provided by the vagus nerve (VN) [3,4,5]. These components are involved in a network of nervous fibers and ganglia that converge in the celiac plexus [5]. Physiologically, VN contains several neurotransmitters, such as acetylcholine, norepinephrine, and dopamine, that are involved in various physiological processes. Acetylcholine is the primary neurotransmitter released by the VN, and it is responsible for many of the VN’s physiological effects on the body’s neurotransmitter receptors, including acetylcholine receptors, adrenergic receptors, and serotonin receptors [6].

The VN is formed by the union of different nerve roots arising from the lateral spinal cord in a groove that forms between the olive and the inferior cerebellar peduncle. It has afferent and efferent functions in the head, neck, chest, and abdomen [7]. The VN is known as cranial nerve X, and it is the only nerve that emerges from the CNS and synapses with ganglia and peripheral organs [8,9]. Vagal fibers found in the medulla oblongata come from four different nuclei in the central nervous system [10]. These fibers converge in a flat cord that passes through the cerebellopontine angle and the cerebellomedullary cistern and leaves the skull through the jugular foramen [11,12].

Vagal influence has a crucial role in homeostasis [13] and is known to regulate the kidneys, adrenal glands, and other abdominal organs that modulate physiological functions associated with these organs [14,15,16]. Additionally, the vagus nerve has been described as a pathway of communication between the ovaries and CNS, specifically with the nuclei of the vagus and the hypothalamus [17]. This connection with the ovaries is provided by the SON and OPN. These create an abdominal network of nerves and ganglia known as the celiac plexus, which is composed of five ganglia: the left celiac ganglion (LCG), right celiac ganglion (RCG), right suprarenal ganglion (RSG), left suprarenal ganglion (LSG) and superior mesenteric ganglion (SMG). These are linked to the lumbar ganglia of the sympathetic trunk (L1–L6) [5]. VN influence on the ovaries has a role in the estrous cycle, pseudopregnancy, the start of puberty, and the secretion of gonadotropins [2,4,18]. Likewise, previous works have shown that vagal activity influences animals with induced polycystic ovary syndrome (PCOS) by modifying the amount of catecholamines, the ovulation rate, and the number of ovulating animals [19,20,21,22]. The specific influence of the VN on the ovaries and how it communicates with them is still unclear. Therefore, in this paper we describe the anatomical pathway of right and left vagal innervation in order to understand VN communication with the ovaries and the neural structures involved in the pathway.

## 2. Materials and Methods

### 2.1. Animals

The experimental protocol was authorized by the Institutional Committee for the Care and Use of laboratory animals, under the specifications of the Mexican Council on Laboratory Animal Care (NOM-062-Z00-1999). All efforts were made to minimize the number of animals used and prevent their suffering during the experiments. This study was performed with eight adult female rats of the Long Evans (CII-ZV) strain (240–300 g), kept under controlled conditions in the Claude Bernard Bioterium of the Benemérita Universidad Autónoma de Puebla (100-310-955-UALVIEP-19/2).

### 2.2. Gross Anatomy

For all procedures, rats were placed on a warmed surgical platform. The animals were anesthetized with an intraperitoneal injection of urethane (ethyl carbamate, i.p. 1.2 g/kg; Sigma-Aldrich, St. Louis, MO, USA) (n = 4). After receiving this lethal dose, the animals survived for approximately two hours. Rats were considered in a surgical plane of anesthesia when the righting reflex was absent, and the toe pinch reflex was negative [23]. All surgical procedures were performed between 9 and 10 h. Subsequently, an incision was made in the ventral position of the abdominal area, removing the skin and muscle, and the left and right vagus nerves were identified in the esophageal area. Caudal nerve pathways were observed under a stereoscopic microscope (Leica M80, Wetzlar, Germany), and the connective and fatty tissue were removed to isolate only the nerves. Schemes and drawings were made, and digital photographs were taken to corroborate the neural organization using a camera (Progres Gryphax^®^ Subra, Jena, Germany) and Gryphax-V1.1.8.153 software coupled to the stereoscopic microscope (Leica M80, Wetzlar, Germany).

### 2.3. Acetylcholinesterase Histochemistry (AChE)

AChE is a common technique that has been used in previous studies to localize very thin nerves or fibers to complement the gross anatomy in describing the peripheral nervous system, because AChE binds cholinergic fibers [1,24,25]. The specific role of acetylcholinesterase is to degrade acetylcholine, ending its signaling activity [26]. In this protocol, animals (n = 4) were anesthetized with sodium pentobarbital, then transcardiac perfusion was performed with Hartmann’s solution (NaCl, NaC_3_H_5_O_3_, CaCl_2_, and KCl) until the tissue was transparent. After perfusion, the thoracolumbar and abdominal regions were dissected under a stereoscopic microscope (Leica M80, Germany), with special care in the paths of the vagus nerves. The fatty and muscular tissue around the nerves and the organs adjacent to the abdominal area of the body were removed to achieve a biochemical reaction that would allow visualizing the distribution of the complete nerve fibers.

Tissue was dissected and fixed in 10% formalin for 48 h. AChE treatment was then carried out and is summarized as follows: in the dark, the tissue was washed in phosphate buffer (1× PBS) and subsequently incubated for 2.5–3.5 h at 40 rpm in a solution of AChE (acetylthiocholine iodide dissolved in 0.1 M sodium hydrogen maleate, 0.1 M sodium citrate, 30 mM CuSO_4_, 5 mM potassium ferricyanide, and 0.1% Triton X-100 (pH 6.0). Next, the tissue was dehydrated and clarified using alcohol and xylol, verifying the staining and dissecting when necessary. Schemes and drawings were made, and digital photographs were taken with the digital camera (Progres Gryphax Subra, Jena Germany, Gryphax software v1.1.8.153, Jena, Germany) attached to the stereo microscope (Leica M80, Wetzlar, Germany).

### 2.4. Histological Analysis

Nissl staining was used to recognize the morphology of neural cells and performed as follows: the tissue was dehydrated in a graded alcohol series of increasing concentration, then treated with distilled water, cresyl violet solution, distilled water, a graded alcohol series of increasing concentration, and xylene.

The procedure for cresyl violet (CAS 10510-54-0; Sigma-Aldrich) staining was similar to that described previously [27,28]. Tissue of the right vagal plexus (RVP) structure localized between the connection of the RVN to the stomach and celiac plexus was removed and further postfixed with Carnoy fixative for 24 h. VP was treated with a graded sucrose series of increasing concentration with changes every 24 h. After this, the RVPs were cut into 10 µm sections with a cryostat (Thermo Scientific, Waltham, MA, USA. 77210163) with a resin of inclusion (Thermo Scientific Shandon Cryochrome, Waltham, MA, USA) at −25 °C.

### 2.5. Immunofluorescence Technique

The protocol was similar to that reported by Bravo-Benitez [29]. The RVPs were cut and preserved with Carnoy fixative for 24 h (n = 4). RVP was treated with a graded sucrose series of increasing concentration with changes every 24 h and cut into 10 µm sections using a cryostat (Thermo Scientific, Waltham, MA, USA. 77210163) with a resin of inclusion (Thermo Scientific Shandon Cryochrome, Waltham, MA, USA) at −25 °C. The sections were defrosted at room temperature and treated for 30 min in 6% fetal bovine serum (FBS; Cat. 26140079, Gibco Thermo Scientific) and 0.1% Tween 20 (Sigma Chemical Co., St. Louis, MO, USA) at 47 °C. Next, they were incubated with the primary antibody for 48 h at 4 °C. Polyclonal anti–rabbit tyrosine hydroxylase (1:500, Cat. AB 152 TH Merck Millipore, Darmstadt, Germany), polyclonal anti–rabbit choline acetyltransferase (1:500 Cat. AB 143 ChAT Merck Millipore, Darmstadt, Germany), and polyclonal anti–rabbit tryptophan hydroxylase 2 (1:500 Cat. ABN 60 TPH Merck Millipore, Darmstadt, Germany) antibodies were used. After incubation with the primary antibody, the tissues were washed with 0.5 M Tris–EDTA buffer (TEB) for 30 s. They were incubated with the secondary anti-rabbit antibody coupled with rhodamine (1:500, Jackson Immuno Research Laboratories, Inc. Baltimore, USA) for 24 h at 4 °C. At the end of this incubation, the tissue was washed with TEB for 30 s. DAPI–Vecta-Shield was added to the tissues (Vector Laboratories, Burlingame, CA, USA) for observation under a fluorescence microscope (Olympus BX41) equipped with an Evolution VF Digital Camera and Image Pro 9.2 software (Media Cybernetics, Bethesda Inc., Rockville, MD, USA). To confirm the correct application of the technique, tissue was used as a negative control in all groups. In this control group, only the primary antibody was omitted, while the remainder of the protocol was continued as directed. Statistical analysis was performed using Abercrombie estimation [30].

## 3. Results

### 3.1. Anatomy of the Right Vagus Nerve

The two vagus nerves in the subdiaphragmatic region were found to differ in their neuroanatomic organization. The right vagus nerve (RVN) is composed of two branches that run together and travel laterally to the esophagus without touching it (Figure 1A). One branch of this structure bifurcates approximately 11 to 14 mm before reaching the lesser curvature of the stomach. This branch further divides and ultimately gives rise to three fibers at the end of the RVN. These fibers converge and form a ganglion-like structure containing numerous fibers and connective tissue (Figure 1B). Due to its anatomical characteristics and origin, this structure has been named the right vagal plexus (RVP). The RVP is located in the space defined by the lesser curvature of the stomach, closer to the cardia region (Figure 1C), and positioned to the right of the esophagus and adjacent to the left gastric artery. From the RVP, one fiber innervates the cranial region of the stomach at the left side, while at right side, another runs laterally towards its insertion in the RCG (Figure 1C,D). No other nervous connections were observed along this pathway. In the abdominal area, various nerve fibers are present, connecting the ganglia of the celiac plexus with other abdominal organs.

Organization of the RVN and LVN in the abdominal region.

### 3.2. Anatomy of the Left Vagus Nerve

In the subdiaphragmatic region, the left vagus nerve (LVN) was observed as an only fiber that runs ventrally to the esophagus (Oe) and travels caudally and attached to it; 2–5 mm before reaching the cranial area of the stomach (st), the LVN bifurcates into two branches. One of these branches inserts into the stomach, and the other one makes a “loop” dorsally to the esophagus and travels laterally in the right direction of the esophagus for 13 to 17 mm until it reaches the RCG where it is inserted into the cranial region of the RCG (Figure 1A). Along this path, no other connections or branches that innervate the LVN were observed (Figure 1B,C).

### 3.3. Right Vagal Plexus as a Center of Synapsis of the RVN

Anatomically, the RVP has characteristics similar to a ganglion, but is smaller in contrast of the ganglia shown of celiac plexus. It is formed with a few somas localized mainly in its center of the structure, many glial cells (Schwann cells, satellite cells), and axons (Figure 2). The enzymatic analysis of the RVP shows that it comprises distal axons, synaptic terminals, axonal organelles that contain biosynthetic precursors, neurons, and glia cells.

The results of the immunoassay conducted on the RVP revealed that it predominantly contains TH encapsulated within synaptic terminals of distal axons. These axons exhibited TH in organelles, forming vesicles of varying sizes distributed throughout the majority of the plexus. However, no neurons reactive to TH were detected within the RVP. Conversely, a small number of ChAT-positive neurons (17 ± 2) and TPH-positive neurons (12 ± 3) were identified in the RVP. Only a few reactive axons were observed for this last two antibodies, localized to a limited peripheral area of the RVP. The negative control specifically targeting TH displayed expected results, while the ChAT and TPH groups exhibited similar outcomes (Figure 3).

The right vagal plexus is located in the abdominal region.

### 3.4. The Vagal Nerve Plexus as a Communicating Network in the Abdominal Area

The VN travels in the thoracic and abdominal zones, innervating tissue and organs (Figure 4A). We identified two different organizations of the RVN. The first has only one branch between the RVP and RCG (Figure 4B, marked with the number 2), while the second has two fibers (Figure 4C, marked with numbers 2 and 3). Similarly, the number of branches between the RCG and the LCG differs between the two compositions, with a single branch in the first arrangement (Figure 4B) and a network of nerves in the second arrangement (Figure 4C). In our study, most individuals (60%) exhibited the first arrangement, while the remaining (40%) displayed the second arrangement.

Both the RCG and LCG are connected to a neural network that links with the enteric system, visceral organs, stomach, adipose tissue, and other abdominal organs. Additionally, our observations corroborate the vagal nervous connection with the prevertebral ganglia through the RCG that innervate the celiac plexus. Otherwise, the ovaries may receive vagal information through the SON, which is connected to the LSG, and the OPN, which joins the SMG, forming a network with the prevertebral ganglia (Figure 4).

Organizations of the RVN and LVN and their connection with ovaries.

## 4. Discussion

The results of this study demonstrate the communication of the vagus nerve with the ovaries, allowing us to understand the relationship between afferent and efferent nervous information. We described the anatomical pathways of both vagus nerves (left and right) in the subdiaphragmatic area and found that they differ. We also observed that both vagus nerves join the celiac plexus specifically in the RCG, and through this ganglion the vagal information travels through the prevertebral ganglia to the SON and the OPN to the ovaries. While the LVN directly innervates the stomach and RCG, the RVN is not a direct route as previously described for the abdominal organs [3,17,31].

As previously described, vagal innervation influences the celiac ganglion [15,32]. However, it is advisable to analyze each prevertebral ganglion separately to know its neural relationship with the right or left vagus nerve. In addition, in the specimens analyzed, no anatomical conjunctions of both nerves were observed proximal to the connection with the RCG. Additionally, the anatomical arrangement of the RVN could prove that is possible that there are multisynaptic signals in the abdominal area, and that the nervous fibers form a network or a single branch connecting nerves and ganglia.

From the existing detailed information on the organization of the celiac plexus [5,25] and what was found in this work, it is evident that the RVN and LVN at the subdiaphragmatic level have different anatomical pathways. It may be possible to analyze the role of each nerve of the ganglia individually, and not as a single signal. In some studies, the prevertebral ganglion is analyzed as a plexus with joint functions known as the celiac superior mesenteric ganglion (CSMG). The CSMG is associated directly with the sympathetic system. On the other hand, the parasympathetic signal is attributed to the vagal signal and carried out to the ovaries [21,25]. However, those studies do not describe the specific anatomical pathway between the VN and gonads. Here, we show that vagal information arrives at the ovaries via an indirect pathway, suggesting that vagal neurons could form synapses in the prevertebral ganglia starting in the RCG, and then through the SON and the OPN to finally reach the gonads. This neural information could be efferent or afferent [33,34,35].

Previous studies [6,15,36,37,38] mention that the VN communicates with the viscera and abdominal organs. This has been corroborated in this work, which adds the fact that this communication involves anatomical structures like the RVP, which is formed by the RVN integrating with distal axons, synaptic terminals, axonal organelles containing biosynthetic precursors, neurons, and glia cells. In addition, we corroborated the observation that right vagal innervation influence on the stomach and the rest of the enteric system is due a nerve fiber that emerges from the RVP and is formed by a network of micro ganglia with mainly catecholaminergic activity. This information also confirms that the RVPs are connected with the nucleus of the origin of the VN in the CNS, where there are catecholaminergic neurons [39].

The existence of this RVP suggests that nerve regulation in the abdominal region involves peripheral control, which could regulate the information from the RVN and influence physiological processes such as digestion, nutrient absorption, and peristalsis, and other physiological functions related to dopamine, epinephrine, and norepinephrine. These neurotransmitters influence the stomach, the ganglia of the celiac plexus, abdominal organs, and the ovaries [36,40]. Our results indicate that the RVP could be formed by distal axons of catecholaminergic neurons, and it may be an area with high synaptic activity and cholinergic and serotoninergic neurons.

In its trajectory, the LVN is a single nerve fiber that bifurcates over the esophagus before its insertion into the stomach. One branch continues its journey towards the RCG and the other continues its way to the stomach. Therefore, both vagus nerves can be considered to communicate with the RCG as the first relay, and through it they connect with the rest of the prevertebral ganglia [5]. Thus, the ovaries receive information from the vagus nerve indirectly.

Previously, VN communication with the ovaries has been observed through labeling with the rabies virus (a transsynaptic tracer) [17]. However, the subdiaphragmatic anatomical route through which this communication takes place was not described. On the basis of the results of our work, we suggest that vagal communication is indirectly related to the ovaries [3,4], and therefore related to structures in the CNS. These include the extrahypothalamic areas (NST, DMV, nucleus ambiguous, area postrema, cell group A7, Barrington nucleus, locus coeruleus, and periaqueductal gray matter) and the dorsal hypothalamus [17], which have neurophysiological roles in ovarian function [18,41,42]. Thus, the VN plays a role in the multisynaptic pathways of communication between the CNS and the ovaries, especially in the synthesis of estradiol and progesterone [43].

## 5. Conclusions

Vagal communication in the abdominal region involves both the RVN and the LVN. The LVN primarily innervates the stomach before reaching the RCG. On the other hand, the RVN pathway includes the RVP, which serves as a connecting link between the stomach and the RCG. The neural composition of the RVP suggests its role as a control center for the transmission of nervous information between the celiac plexus, the stomach, and the CNS. From the RCG, connections are established with the enteric system and organs including both ovaries, through the synaptic connections of neurons within the prevertebral ganglia of the celiac plexus. These findings indicate that vagal innervation may play a role in the metabolism of the ovaries, potentially serving as both an afferent and efferent pathway. However, it is important to note that there is no direct pathway connecting the gonads and the vagal nuclei in the CNS.

## Figures and Tables

**Figure 1 brainsci-13-01032-f001:**
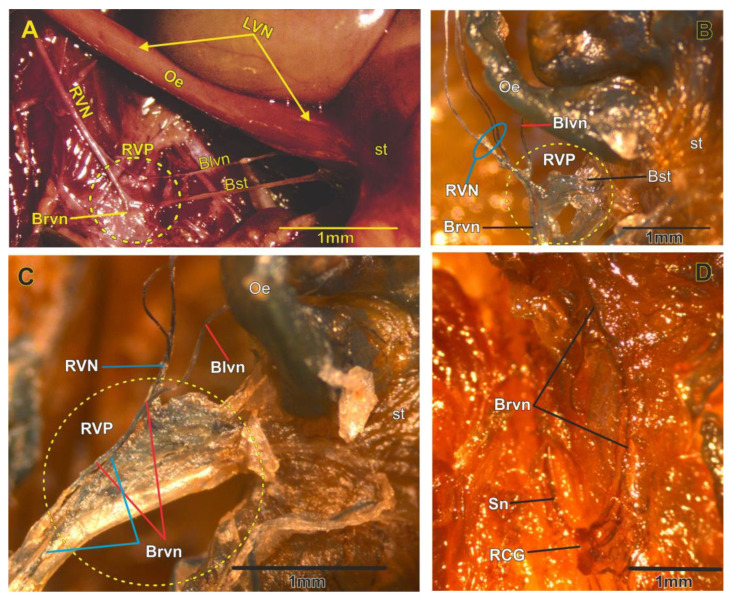
Panel of photographs that show the organization of the RVN and LVN. (**A**) The LVN travels on the esophagus (Oe) to the stomach (st); it bifurcates before reaching the stomach, where a nerve branch continues towards the stomach and another makes a turn in the posterior area of the esophagus to continue towards the RCG (Blvn). The RVN is divided to form three nervous fibers (**B**) that converge in one vagal plexus formed by microganglia (right vagal plexus [RVP] (**C**) in the yellow circle) not described before. From the RVP emerge two fibers: one of them travels to the stomach (Bst) and the other goes to the RCG (Brvn). In the abdominal area is observed the connection of LVN, RVN that reach the RCG with other branches and ganglia in celiac plexus (**D**). Sn, splenic nerve. Gross anatomy (**A**) and AChE (**B**–**D**).

**Figure 2 brainsci-13-01032-f002:**
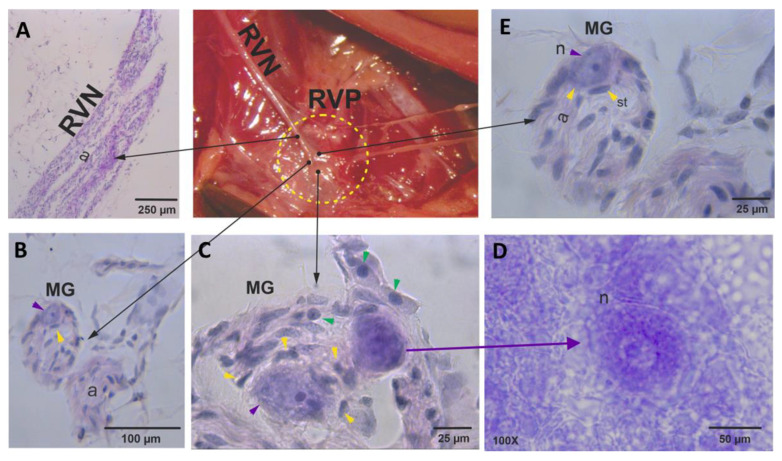
Histological analysis of the right vagal plexus (RVP). (**A**) View of axons that run to the RCG and correspond to the RVN. (**B**) The RVP is formed by a network of microganglia (MG) (**E**) and vesicles at the end of axons from the RVN (a). Additionally, we observed Schwann cells (green arrows) and satellite cells (st, yellow arrows) (**C**), and the soma of neurons (n, purple arrow) (**D**). Nissl stain.

**Figure 3 brainsci-13-01032-f003:**
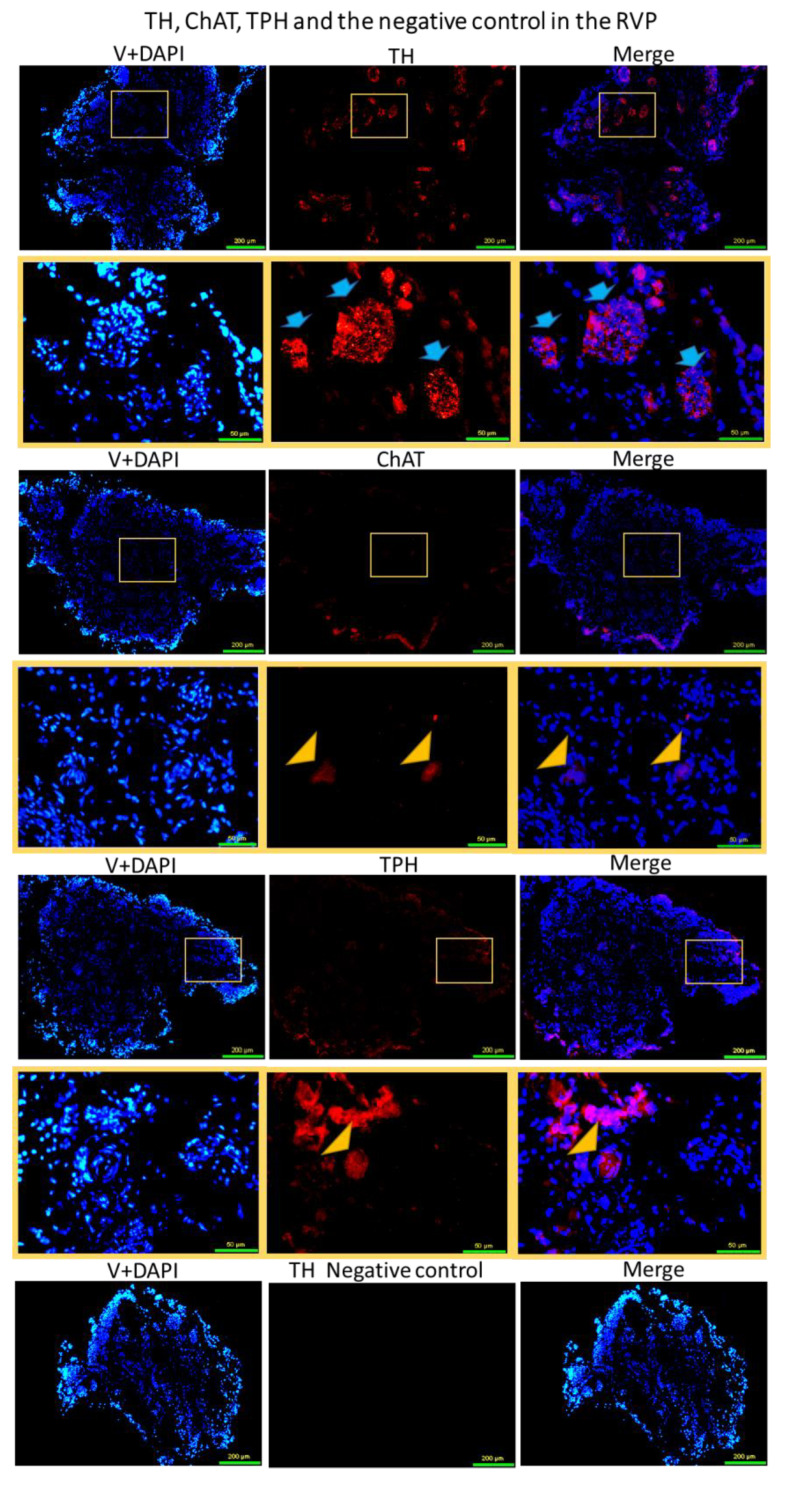
Immunofluorescent staining for TH, ChAT, and TPH, and the negative control, in the RVP. TH is localized in almost all regions of the RVP. There are many distal axonal organelles of different sizes and shapes (blue arrows) that contain TH, and it is the most observed structure in this plexus. Some ChAT-positive and TPH-positive neurons (marked with yellow triangle) are localized in the RVP, and only axonal reactivity is observed in the periphery of the ganglia for the last two antibodies. The yellow box zoom in the image below is highlighted for each image. Scale bar 200 μm and close up 50 μm.

**Figure 4 brainsci-13-01032-f004:**
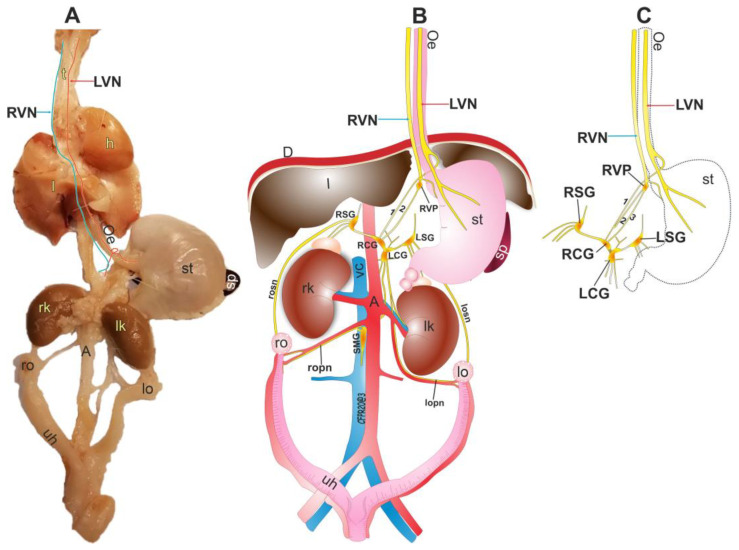
Organization of the subdiaphragmatic vagus nerves. (**A**) Photography of the ventral view of the organs and the RVN and LVN. (**B**) Scheme that shows the organization of the LVN and RVN with the prevertebral ganglia and ovaries. The right vagal organization with only two branches between the RVN and RCG was observed in most animals. (**C**) Scheme that shows the second right vagal organization with three branches. RVN, right vagus nerve; LVN, left vagus nerve; h, heart; l, lungs; Oe, esophagus; st, stomach; rk, right kidney; lk, left kidney; A, aorta; ro, right ovary; lo, left ovary; uh, uterus; l, liver; D, diaphragm; RSG, right suprarenal ganglion; LSG, left suprarenal ganglion; RCG, right celiac ganglia; LCG, left celiac ganglia; SMG, superior mesenteric ganglia; RVP, right vagal plexus; CV, cava vein; rson, right superior ovarian nerve; lson, left superior ovarian nerve; ropn, right ovarian plexus nerve; lopn, left ovarian plexus nerve.

## Data Availability

Not applicable.

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
