# Peer review of "Organization of the Subdiaphragmatic Vagus Nerve and Its Connection with the Celiac Plexus and the Ovaries in the Female Rat"

_brainsci, 2023, doi:10.3390/brainsci13071032_

Round 1

Reviewer 1 Report

The manuscript is good written, findings are very interesting and good documented.
I recommend this paper to publish in the Brain Sciences.

Author Response

Observation

Comments

R1.

Improve the methods description

The methods were reviewed and supplemented in accordance with the published protocols for describing certain nerves and ganglia of the peripheral nervous system (References 1, 5, 23, and 24).

R1.

Improve the results presented

The results were reviewed, and the sections were refined and expanded to ensure a clearer presentation of the information. All sections of the results, including the figure captions, were carefully reviewed.

R1.

Conclusions supported by the results

The conclusions section was modified to provide a detailed explanation of the findings obtained in this study.

R1. Comments and suggestions

The manuscript is good written, findings are very interesting and good document.

 Thank you

Reviewer 2 Report

The authors investigated the neuroanatomical connections between the ovaries and the vagus nerve via gross anatomy and histological analysis. This study provides innovative findings, and the manuscript is well-prepared. However, here are some suggestions for improving the draft.

-       Please explain why TPH 2 instead of TPH 1 has been used in this study.

-       Did the authors conduct any quantifications for the IHC results?

-       Line 81 (“in the area the esophagus”) is not clearly stated. Please revise.

It is adequate for publication. 

Author Response

Observation

Comments

R2.

Minor editing of English language

The manuscript was revised, and corrections were made in all sections.

R2.

Study design 

The study design was conducted based on previous references that provided neuroanatomical descriptions of the peripheral nervous system (References 1, 5, 23, and 24). The materials, animals, and antibodies were carefully selected and utilized under appropriate conditions to ensure reliable results.

R2.

Improve the methods description

The methods were reviewed and enhanced based on published protocols for describing specific nerves and ganglia of the peripheral nervous system (References 1, 5, 23, and 24).

R2. Comments and suggestions

The authors investigated the neuroanatomical connections between the ovaries and the vagus nerve via gross anatomy and histological analysis. This study provides innovative findings, and the manuscript is well-prepared.

Thank you

R2.

Please explain why TPH2 instead of TPH 1 has been used in this study.

According to the literature, Tryptophan hydroxylase 2 (TPH2) catalyzes the initial and rate-limiting step in serotonin biosynthesis. In comparison to TPH1, TPH2 exhibits several advantageous characteristics. Firstly, TPH2 is more soluble, which enhances its stability and availability within the peripheral nervous system. Additionally, TPH2 has a higher molecular weight, making it less prone to degradation and more likely to remain intact during experimental procedures. Moreover, TPH2 possesses distinct kinetic properties, including a lower catalytic efficiency towards phenylalanine compared to TPH1. This specificity allows for a more precise targeting of serotonin synthesis in the peripheral nervous system. Taking these factors into account, we made the decision to use the TPH2 antibody as it provides a greater assurance of correct adherence to the peripheral nervous system and increased reliability in identifying nerve activity within the right vagal plexus (RVP).

References

DOI: 10.1002/2211-5463.12100

DOI: 10.1111/j.1471-4159.2004.02850.x

R2.

Did the authors conduct any quantifications for the IHC results?

The results of the "Right vagal plexus as a center of synapsis of the RVN" section were modified to include the number of ChAT-positive and TPH-positive neurons observed in the RVP. The Abercrombie estimation method was employed to accurately count these neurons.

DOI.org/10.1002/ar.1090940210

R2.

Line 81 (“in the area the esophagus”) is not clearly stated. Please revise

The "Gross anatomy" section, specifically at line 81, was revised, and the wording was modified to better describe the method employed in this particular part of the study.

Reviewer 3 Report

In this paper, the authors try to demonstrate by using anatomical and immunohistochemical approaches the communication of the vagus nerve with the ovaries.

The results are in complex clear and the different techniques produce compatible results and conclusions.

However, I strongly recommend improving the quality of immunofluorescence images, in particular, that for TpH. Moreover, there is a very bad background signal around the sections.

The authors must specify the type of negative control: for example omission of primary antibody or other approach.

Minor editing of English language is required.

Author Response

Observation

Comments

R3.

Minor editing of English language

The manuscript underwent a thorough revision, and various corrections were made across all sections.

R3.

Study design 

The study design was based on previous references that provided neuroanatomical descriptions of the peripheral nervous system (References 1, 5, 23, and 24). Careful selection of materials, animals, and antibodies was carried out to ensure reliable results, and appropriate conditions were maintained throughout the experimental procedures.

R3.

Improve the methods description

The methods description were thoroughly reviewed and supplemented with protocols published previously (1, 5, 23, and 24) to accurately describe specific nerves and ganglia of the peripheral nervous system.

R3.

Improve the results presented

The results underwent a comprehensive review process, during which the sections were enhanced and expanded to ensure a clearer and more effective presentation of the information. All sections of the results, including the figure captions, were carefully reviewed to enhance their clarity and coherence.

R3.

Conclusions supported by the results

The conclusions section was modified to provide a detailed explanation of the findings obtained in this study. It was revised to offer a comprehensive and in-depth analysis of the results, allowing for a more thorough understanding and interpretation of the study's outcomes.

R3. Comments and suggestions

In this paper, the authors try to demonstrate by using anatomical and immunohistochemical approaches the communication of the vagus nerve with the ovaries. The results are in complex clear and the different techniques produce compatible results and conclusions.

Thank you

R3.

Recommendation to improve the quality of immunofluorescence images.

We enhanced the quality of the images by removing the background from all of them. Additionally, we replaced the previous TPH close-up image with a higher-quality image that clearly depicts the TPH-positive neuron. These improvements were made to ensure better visual clarity and to enhance the overall quality of the images presented in the study.

R3.

Specify the type of negative control

In the methods section, the Immunofluorescence technique was revised and supplemented with information regarding the negative control. It was specified that the negative control group did not receive the first antibody during the staining process.